# Factors Associated with Decisions for Initial Dosing, Up-Titration of Propiverine and Treatment Outcomes in Overactive Bladder Syndrome Patients in a Non-Interventional Setting

**DOI:** 10.3390/jcm10020311

**Published:** 2021-01-15

**Authors:** Marjan Amiri, Tim Schneider, Matthias Oelke, Sandra Murgas, Martin C. Michel

**Affiliations:** 1Institute of Medical Informatics, Biometry and Epidemiology, University Hospital Essen, 45130 Essen, Germany; marjan.amiri@uk-essen.de; 2Center for Clinical Trials Essen (ZKSE), University Hospital Essen, 45130 Essen, Germany; 3Praxisklinik Urologie Rhein-Ruhr, 45468 Mülheim, Germany; t.schneider@pur-r.de; 4Department of Urology, St. Antonius Hospital, 48599 Gronau, Germany; matthias.oelke@st-antonius-gronau.de; 5Apogepha, 01309 Dresden, Germany; s.murgas@apogepha.de; 6Department of Pharmacology, Johannes Gutenberg University, 55131 Mainz, Germany

**Keywords:** propiverine, dose-titration, overactive bladder syndrome, allocation bias, escalation bias

## Abstract

Two doses of propiverine ER (30 and 45 mg/d) are available for the treatment of overactive bladder (OAB) syndrome. We have explored factors associated with the initial dosing choice (allocation bias), the decision to adapt dosing (escalation bias) and how dosing relative to other factors affects treatment outcomes. Data from two non-interventional studies of 1335 and 745 OAB patients, respectively, receiving treatment with propiverine, were analyzed post-hoc. Multivariate analysis was applied to identify factors associated with dosing decisions and treatment outcomes. Several parameters were associated with dose choice, escalation to higher dose or treatment outcomes, but only few exhibited a consistent association across both studies. These were younger age for initial dose choice and basal number of urgency and change in incontinence episodes for up-titration. Treatment outcome (difference between values at 12 weeks vs. baseline) for each OAB system was strongly driven by the respective baseline value, whereas no other parameter exhibited a consistent association. Patients starting on the 30 mg dose and escalating to 45 mg after 4 weeks had outcomes comparable with those staying on a starting dose of 30 or 45 mg. We conclude that dose escalation after 4 weeks brings OAB patients with an initially limited improvement to a level seen in initially good responders. Analysis of underlying factors yielded surprisingly little consistent insight.

## 1. Introduction

Muscarinic receptor antagonists (antimuscarinics) are the mainstay of treatment of the overactive bladder (OAB) syndrome [1,2]. Several of the clinically available members of this drug class are available in multiple dose strengths, for instance darifenacin [3], fesoterodine [4], propiverine [5] and solifenacin [6]. This allows increasing the dose in patients with an insufficient improvement of symptoms and decreasing it in those with bothersome adverse drug reactions (ADRs). While it appears intuitive to assume that a greater dose will have greater effects, clinical experience demonstrates that this is not necessarily the case. This is expected because the law of mass action postulates that a greater dose of antagonist/inhibitor will have a greater effect only until a ceiling limit is reached due to maximum occupancy of the molecular target of the drug; for instance, dose escalation of the antidepressant paroxetine has been shown not to improve symptoms because the standard dose already occupies most of the serotonin transporters [7]. On the other hand, the optimal position within the dose–response curve of a given drug, i.e., where an optimal ratio between efficacy and tolerability is achieved, can differ between patients [8]. Prediction of the optimal dose for a given patient is difficult because factors, such as age and comorbidity [9], genotype [10] or, for antimuscarinics, concomitant medication also acting on muscarinic receptors [11], may pharmacodynamically affect the drug response. Moreover, depending on the specific metabolic pathways responsible for the elimination of a given antimuscarinics, pharmacokinetic factors, including age, renal function and genotype of drug-metabolizing enzymes, may also play a role [12].

The prescribing information of most antimuscarinics for the treatment of OAB defines a recommended starting dose with the option to increase the dose if tolerability is acceptable and greater efficacy is needed. Accordingly, the effect of dose escalation has been studied for several antimuscarinics including fesoterodine [13,14,15,16,17] and solifenacin [18,19]. A variation of this theme has been to study the effects of escalating from a low dose of one drug to the standard dose of another one within the same drug class [20] or to model the effects of dose escalation based on findings from previous trials [21]. These studies demonstrate that the decision to increase the dose was often but not always associated with the extent of improvement of OAB symptoms during the initial treatment period; however, which OAB symptoms were primarily associated with that decision differed considerably between studies. Some reports also proposed that baseline symptom severity, body weight and gender (based social/cultural roles and personal identity) may be associated with the decision for dose escalation [15], but this was not found consistently. Interestingly, the degree of symptom improvement in the initial treatment period has also been found to be associated with the decision to increase dosage within the placebo arm of controlled dose-escalation studies [22].

Propiverine is an antimuscarinic for the treatment of the OAB syndrome that differs from other members of this drug class because the compound and some of its metabolites additionally have inhibitory effects on L-type Ca^2+^-channels [23,24], which play a role in the control of bladder smooth muscle tone [25]. Moreover, propiverine extended release (ER) has two approved starting doses of 30 and 45 mg once daily, i.e., the higher dose is not only available as part of dose escalation. As most other antimuscarinics have only one approved starting dose, it has not been reported, to our knowledge, which factors are associated with the initial dosing decision. Moreover, it is possible that factors associated with the initial dosing decision also impact the decision to increase the dose and, thereby, the overall treatment outcome. Against this background, we report two non-interventional studies (NISs) of similar design in which patients with OAB syndrome were treated with either dose of propiverine for a planned observation period of approximately12 weeks and the possibility for dose-adjustment after about 4 weeks. While reporting on the primary outcomes of both studies, we have also applied multivariate analysis (general linear models) in a post-hoc approach to explore three questions:

Which factors are associated with initial dosing decision (allocation bias)?

Which factors are associated with a decision for dose escalation after 4 weeks of treatment (escalation bias)?

How much of differential efficacy of the two dose strengths can be attributed to greater dose and how much to other factors associated with dosing choice?

## 2. Materials and Methods

Two NISs with a similar design were performed. Study I was primarily designed to explore the effects of different starting doses and dose adjustment after 4 weeks with a treatment duration of 12 weeks, i.e., specifically designed for the purpose of the present analyses. Study II was designed to explore the effect of additional material (information sheet about OAB and mode of action of the drug) on efficacy and tolerability and on premature discontinuation during a treatment period of 12 weeks and allowed extension of observation for up to 24 weeks in a subgroup. Otherwise, its design was very similar to that of study I. Therefore, we have used both studies to address our research questions. Our exploratory approach considers study I as hypothesis-generating and study II as exploring the robustness of the findings from study I but not as formally hypothesis-testing. To keep the manuscript readable, data related to the primary aim of study II and some other outcomes are shown in the Appendix A. We have previously used parts of both datasets to explore the presence or absence of a normal distribution of the OAB parameters urgency, incontinence, frequency and nocturia and of treatment-associated alterations thereof [26]. We now report the primary analysis of efficacy and tolerability of the full dataset of both studies. Both studies were based on §67, 3 of the German Drug Act and had been approved by the ethical committee of the state board of physicians in Saxony, Germany (Sächsische Landesärztekammer EK-BR-14/12-1 and EK-BR-18/14-1).

Based on their non-interventional character, both studies lacked formal inclusion or exclusion criteria other than the Summary of Product Characteristics. Rather, 456 and 158 participating physicians in studies I and II, respectively, were asked to systematically document their observations in patients who were to be treated with propiverine ER (30 or 45 mg once daily) based on the physician’s medical judgment. Three visits were planned during the observation period, visit 1—one at baseline (initiation of propiverine prescription and selection of starting dose), visit 2—after about 4 weeks (possibility of dose adaptation) and visit 3—after about 12 weeks (study end); study II allowed us to extend the observational period to 24 weeks. The first visit of the first patient was recorded on 1.7.2010 and the last visit of the last patient was on 23.1.2013 in study I and on 14.1.2014 and 15.7.2015 in study II.

During each visit, parameters related to OAB and decisions on subsequent dosing were recorded. Demographics, comorbidities and comedications were additionally recorded at baseline. Global tolerability (rated as very good, good, sufficient or insufficient) was assessed by the patient and the physician at visits 2 and 3. Data on ADR were collected throughout the entire study. Post-void residuals (PVRs) were considered as additional safety parameter if available (recording not mandatory to maintain the non-interventional character of the studies). The safety and tolerability analysis included all patients who had taken at least one drug dose and had at least one physician contact thereafter. The efficacy analyses included all patients who had OAB-related data at baseline and at least one time point thereafter.

In line with the non-interventional character of the study, the protocol did not specify whether OAB-related data were collected from voiding diaries or from patient recollection, but the applicable German guideline at the time the studies were performed recommended recording of voiding diaries [27]. Categorical data (e.g., gender or dose) are shown as % of the respective population. Continuous data are expressed as means ± SD (age, height, weight, body mass index (BMI)) or as median with inter-quartile range (IQR (reported as lower and upper quartile separated by a “;”); OAB duration and daily episodes of urgency, incontinence, voids and nocturia), depending on whether variability was considered to exhibit a normal distribution [26]. Despite deviating from a normal distribution, OAB parameters are also shown as means ± SD to facilitate comparison with previously reported studies. Patients with medically implausible values (urgency > 50, incontinence > 30, frequency > 40, and nocturia > 20 episodes/24 h) were excluded from the analysis for that symptom and visit; this affected four patients each for urgency and frequency, one each for incontinence and nocturia in study I and none in study II.

Data handling and statistical analyses were performed by Bioconsult GmbH (Rickenbach, Switzerland), a contract research organization, based on a statistical analysis plan developed by the authors and using SAS version 9.4 (SAS Institute, Cary, NC, USA). The demographic data and OAB parameters at baseline and subsequent visits for the efficacy population are shown descriptively. Univariate analyses compared (a) demographics and baseline data in patients starting at the 30 mg and 45 mg dose at visit 1, (b) such data plus initial OAB symptom changes at visit 2 in patients starting at 30 mg and either staying on that dose or increasing it to 45 mg, and (c) treatment outcomes at visit 3 for patients who started and stayed on 30 mg, who started on 30 mg and increased to 45 mg and who started on 45 mg and stayed on that dose. These data were analyzed for descriptive *p*-values using the Kruskal–Wallis test.

Multivariate analyses (general linear models) were applied to identify factors associated with initial dosing decisions and dose escalation decision as well as the relative roles of dose and other factors in treatment outcomes. Variables were incorporated into the model if *p* < 0.03 and removed if *p* > 0.35 in the next step; the procedures were stopped when only variables with *p* < 0.35 were retained in the model. The model for the initial dosing choice included gender, age, body weight, height, BMI, duration of OAB and baseline values for urgency, incontinence, frequency and nocturia as potential explanatory variables. Precision of parameter estimates in the general linear models is indicated by their standard error (SE). Treatment-associated changes of a symptom were determined only for patients exhibiting that symptom at baseline, e.g., for incontinence episodes only in those having ≥ 1 incontinence episode; however, not exhibiting one symptom at baseline did not preclude analyses of other symptoms of the same patient. The model for factors associated with dose escalation included the same variables and additionally the OAB parameters after 4 weeks at visit 2. The model for treatment outcomes also included the same variables and additionally the dose after visits 1 and 2.

Based on the exploratory character of the study and in line with recent recommendations [28], no hypothesis-testing statistical analysis was applied. Therefore, all reported *p*-values should be interpreted as descriptive only. Rather, we have considered study I in line with its primary aim as hypothesis-generating and study II to check for robustness of our findings. Overall reporting follows the STROBE guidelines for cohort studies (https://strobe-statement.org).

## 3. Results

### 3.1. Patient Flow and Baseline Data

A total of 1335 and 745 patients participated in studies I and II, respectively. Demographics of both studies, baseline symptoms, documented prior interventions with implications for lower urinary tract function, comorbidities and comedications, as well as overall patient flow, are shown in the Appendix A.

### 3.2. Descriptive Analysis of Treatment Outcomes

While no specific instructions on time of administration were given, most patients in study I (919/1120; 82.1%) reported taking propiverine in the morning. After 4 weeks of treatment, clinically meaningful improvements (median with IQR and mean ± SD in parentheses) were observed in the overall efficacy population with a reduction in urgency episodes by 4 (2; 7; 5.1 ± 4.6), in incontinence episodes by 2 (1; 4; 2.8 ± 3.1), in micturitions by 4 (2; 6; 4.2 ± 3.3), and in nocturia episodes by 1 (1; 2; 1.5 ± 1.3). Between weeks 4 and 12, the overall cohort of patients reported additional improvements in urgency episodes by 1 (0; 3; 1.8 ± 2.6), in incontinence episodes by 1 (0; 1; 0.9 ± 1.6), in frequency by 1 (0; 2; 1.4 ± 1.9), and in nocturnal voids by 0 (0; 1; 0.5 ± 0.9). Thus, overall improvements from baseline to week 12 were improvements of urgency episodes by 6 (3; 10; 6.9 ± 5.2), in incontinence episodes by 3 (1; 5; 3.7 ± 3.3), in frequency by 5 (3; 8; 5.7 ± 3.7), and in nocturnal voids by 2 (1; 3; 2.0 ± 1.4). Treatment effects in the various groups are summarized in Figure 1 based on medians and IQR and in the Appendix A for means ± SD. Corresponding data for study II were comparable and are shown in the Appendix A.

### 3.3. Factors Associated with Dosing Decision at Visit 1

Demographic and OAB-related baseline data of patients initially receiving 30 or 45 mg propiverine are summarized in Table 1 for study I. The two groups had a very similar gender distribution, mean age and height, but those starting on 45 mg had slightly greater mean body weight (+2.9 kg). The 45 mg group also had greater baseline symptom severity, which was most notable for number of incontinence episodes and micturitions with medians being 1–2 episodes greater than in the 30 mg group. Data for study II were comparable and are shown in the Appendix A.

After exclusion of subjects due to missing values for the response or explanatory variables, the logistic regression analysis included 606 and 147 patients from study I, starting at the 30 and 45 mg dose, respectively, and 294 and 132 subjects from study II. A younger age was the only variable associated with a higher starting dose in both studies with a *p* < 0.05. Additionally, both a greater basal number of incontinence and nocturia episodes were associated with a higher starting dose in study I and a longer duration of OAB and greater number of micturitions in study II (Table 2).

### 3.4. Factors Associated with Dosing Increase at Visit 2

The dose of propiverine was adjusted in some patients continuing treatment at visit 2 (patient disposition figures for studies I and II are shown in the Appendix A). In study I, 84% of patients having started with a dose of 30 mg remained on that dose (30/30 group), whereas 16% were switched to the 45-mg dose (30/45 group). In addition, 88% of patients having started with a dose of 45 mg remained on that dose (45/45 group), and 12% reduced it to 30 mg; the latter group was not considered further as it was deemed too small to allow meaningful analysis. Corresponding data for study II are shown in the Appendix A.

Table 3 shows demographic and OAB-related data at baseline and after 4 weeks of treatment in the 30/30 and the 30/45 group of study I. Patients with dose escalation were slightly taller (2.3 cm) and heavier (4.2 kg). Prior to the start of treatment, they had more daily urgency, incontinence, micturition and nocturia episodes. These differences were maintained after 4 weeks of treatment and became even greater except for nocturia. Corresponding data for study II are shown in the Appendix A.

After exclusion of subjects due to missing values for the response or explanatory variables, the logistic regression analysis included 834 and 161 patients from study I staying at the 30 mg dose or increasing to 45 mg, respectively, and 435 and 59 subjects from study II. A greater number of urgency episodes at baseline and a greater number of incontinence episodes after 4 weeks of treatment were associated with dose escalation in both studies with a *p* < 0.05; however, greater height was associated only in study I and greater number of nocturia episodes were only associated in study II (Table 4).

### 3.5. Factors Associated with Treatment Outcomes

As it cannot necessarily be assumed that factors associated with improvement of one symptom are the same as for other symptoms, we have explored factors associated with improvement (greater improvement = small symptom episode frequency) separately for each OAB parameter with the delta between value after 12 weeks of treatment as dependent and respective basal value as independent variable. These logistic regression analyses considered the 30/30, 30/45 and 45/45 groups. Demographics and baseline values for the 30/30 and 30/45 groups in study I are shown in Table 3 and those for the 45/45 group in Table 1. Values of OAB parameters after 4 weeks of treatment are shown in Table 3 for the 30/30 and the 30/45 group; for the 45/45 group, they were 5 (5.1 ± 3.9) for urgency, 2 (2.6 ± 3.1) for incontinence, 9 (9.5 ± 3.3) for frequency and 2 (2.1 ± 1.2) for nocturia (corresponding data for study II are shown in the Appendix A).

After exclusion of subjects due to missing values for the response or explanatory variables, the logistic regression analysis included 699 patients from study I and 396 from study II. The only variable consistently associated with treatment outcome for any OAB symptom was the baseline value of the same symptom, i.e., baseline urgency for overall improvement of urgency (Table 5) and baseline incontinence for overall improvement of incontinence (Table 6); logistic regression results for frequency and nocturia are shown in the Appendix A. Compared to the 45/45 group, being in the 30/30 or 30/45 group was associated with smaller treatment-associated improvements for urgency and incontinence in study I, but not in study II (Table 5 and Table 6 and Appendix A); dosing regimen had no statistically significant effects on treatment-associated improvements of frequency and nocturia in either study (Appendix A).

### 3.6. Safety and Tolerability

Safety and tolerability were assessed in three ways: Firstly, PVR was unchanged (study I: 10 (10; 30) ml at baseline and after 12 weeks of treatment; study II: 20 (0; 38) ml at baseline and 20 (0; 39.5) ml after 12 weeks of treatment). Urinary retention was reported for no patient in study I and one patient in study II.

Second, 324 patients (24.3%) in study I reported a total of 461 ADR with dry mouth (19.6%) and constipation (6.0%) mentioned most frequently. The incidence of ADR was comparable between dose regimens. A total of 145 patients (10.9%) discontinued treatment during or after the treatment period in study I (48 due to ADR, 38 insufficient efficacy, 33 based on patient wish, 14 because of being symptom-free, 8 due to other and 3 due to unknown reasons). This included 71 patients up to visit 2 with similar incidence with a starting dose of 30 and 45 mg (57/1069, 5.3% vs. 13/249, 5.2%). It also included 74 patients up to visit 3 (41/875, 4.7% received the 30 mg dose and 33/371, 8.9% the 45 mg dose). Reasons for discontinuation were ADR (*n* = 48), insufficient efficacy (*n* = 38), patient wish (*n* = 33), having become symptom free (*n* = 14), and other reasons (*n* = 12); no information on the reason for discontinuation was recorded for three patients. Patients with discontinuation of treatment after the planned observation period of 12 weeks were included in the calculations of treatment effects.

In study II, 163 patients (21.9%) reported 231 ADR with dry mouth (16.1%) and constipation (5.5%) mentioned most frequently and mostly being rated as mild. The capture of treatment discontinuation at visit 3 was not comparable with study I, because of the voluntary extension after 12 weeks. At visit 3, treatment discontinuation was observed in a total of 83/745 patients (11.1%), 5.2% due to ADRs, 1.9% insufficient efficacy, 0.9% based on patient’s own decision, 0.8% because of being symptom-free and 0.8% due to other reasons. The incidence at visit 3 regarding the starting dose was similar (30 mg: 62/531, 11.7% vs. 18/200, 9.0%).

Third, global tolerability was rated by the patient at study end as very good, good, sufficient or insufficient in 41.3%, 47.2%, 10.3% and 1.3% for propiverine 30 mg and in 38.6%, 49.6%, 9.6% and 2.2% for propiverine 45 mg in study I. In study II, the rating for propiverine 30 mg was 43.0%, 45.7%, 7.6% and 3.7% for very good, good, sufficient or insufficient and correspondingly for propiverine 45 mg 36.3%, 54.0%, 6.8% and 3.0% at study end.

## 4. Discussion

The present analyses attempted to address three questions:

Which factors are associated with initial dosing decision (allocation bias)?

Which factors are associated with a decision for dose escalation after 4 weeks of treatment (escalation bias)?

How much of differential efficacy of the two dose-strengths can be attributed to greater dose and how much to factors associated with the dosing decision?

While the second question had previously been addressed in several studies with other muscarinic antagonists, the first and third question, to the best of our knowledge, are addressed here for the first time, most likely because antimuscarinics other than propiverine do not allow a choice of starting doses according to their respective prescribing information.

### 4.1. Critique of Methods

NISs differ from randomized, controlled trials (RCTs) in several ways. As NISs typically lack a placebo or other comparator group, they do not allow direct conclusions on the efficacy or tolerability of a drug; however, they describe what realistically can be expected to occur in routine clinical practice. For instance, the treatment of lower urinary tract symptoms (LUTSs) is known to exhibit a strong placebo effect [29]. Accordingly, antimuscarinics as a class have little effect on nocturia relative to placebo in RCT [30], but improvements of nocturia upon treatment with antimuscarinics in OAB patients [31,32] or α_1_-adrenoceptor antagonists in male LUTS patients [33,34] have consistently been reported from NISs, as also observed in the present study. On the other hand, NISs lack the strict inclusion and exclusion criteria typical for RCTs, which leads to a less artificial study population. Thus, RCTs have a greater internal, whereas NISs can have a greater external validity.

In line with their non-interventional character, NISs have fewer rules on how data are to be captured than RCTs. While the applicable German guideline at the time the studies were performed recommended assessing OAB symptoms based on voiding diaries [27], we have no data regarding whether this has been implemented consistently.

Our analyses are based on two NISs that had a very similar design. There were three differences: Study II formally had a distinct primary aim (effect on information material), had an option to extend the observation to 24 weeks, and was performed later in time (2014–2015 as compared to 2010–2013). With regard to the latter, it is important to note that no major changes had occurred between the two time frames in the overall German healthcare system, or in general, regarding recommendations related to the treatment of OAB syndrome. Thus, we did not identify major differences in treatment approaches or overall treatment outcomes and had expected to obtain very similar findings in both studies. Nonetheless, we decided not to pool the data but rather to analyze them in parallel. The main reason was that this would enable us to see whether analytical outcomes would be consistent in two distinct samples of the OAB syndrome population.

Our previous systematic review of the literature has demonstrated that the majority of studies are reporting means ± SD of OAB symptoms; while this implies the assumption of a normal distribution of these parameters, almost all studies failed to provide evidence in this regard [26]. On the other hand, our own analyses of the databases underlying the present manuscript found that distribution of OAB symptoms clearly deviates from normality and that means and medians differ systematically [26]. Therefore, we primarily report medians and IQR for OAB symptoms and have applied non-parametric statistical tests (which do not assume a normal distribution); however, to facilitate comparison with published data from other studies, we also report means ± SD.

The prevalence of reported comorbidities and comedications in the two studies reported here is higher than in previously reported NISs in OAB syndrome from Germany with propiverine [35] or other antimuscarinics such as darifenacin [32], fesoterodine [36], solifenacin [37] or tolterodine [31]. Whether this reflects a more careful documentation by physicians participating in the present study or a shift in patient populations over time cannot be determined with certainty based on the present data. However, overall improvements in symptoms were comparable in the present and previous NIS with muscarinic antagonists in OAB patients, confirming the conclusion from the previous meta-analysis of the RCT with these drugs [2]; comparable improvements have also been reported from a NIS with propiverine in male LUTS [35]. All of these aspects should be kept in mind in the interpretation of the present data.

### 4.2. Factors Associated with Initial Dosing

Propiverine ER differs from other muscarinic antagonists in having two approved starting doses. This allowed us to explore factors associated with starting dose selection by participating physicians, a question not addressed in any previous study. While subjects starting with the 45 mg dose had a slightly greater body weight and symptom severity, our logistic regression analyses identified age as the only variable consistently associated with the choice of starting dose. Thus, younger patients were more likely to start at the 45 mg. This may reflect the idea that young subjects may be less vulnerable to ADR of antimuscarinics [38]. On the other hand, factors such as duration of OAB syndrome, number of incontinence, voiding and nocturia episodes were associated with starting dose in one, but not the other, of the two studies. Moreover, all associations of these explanatory variables with the starting dose were of moderate strength only. This highlights the importance of testing for consistency of findings across multiple databases.

### 4.3. Factors Associated with Dose Escalation

Several previous studies have explored factors associated with a decision for dose escalation during treatment. This included both RCTs [15,17,18] and non-randomized studies [13,14,19]. In the present analyses, the number of urgency episodes at baseline and the change in incontinence episodes after 4 weeks of treatment were the only parameters consistently associated with the decision to increase the dose from 30 to 45 mg. In contrast, height, basal number of nocturia episodes and change thereof were associated with dose escalation in one, but not the other, study. Greater baseline symptoms, although not necessarily nocturia as in the present studies, had also been reported to be associated with a decision for dose escalation in some previous studies with other antimuscarinics [13,17,18], but this was not confirmed in others [15]. Similarly, smaller OAB symptom improvements in the initial treatment period were associated with a decision to increase the dose in some studies [13,14,18] but not in others [15,17]. Adding to this heterogeneity in findings, dose escalation was associated with greater baseline values and/or smaller initial improvements sometimes for all OAB symptoms, but sometimes such as also observed in the present study for only selected OAB symptoms. Interestingly, a smaller initial treatment response was also reported to be associated with sham dose escalation within the placebo arm of RCT [22]. An inconsistency of reported associations with dose escalation also applies to factors other than OAB symptoms, including age ([14] and present studies), BMI ([14,15] and present studies), OAB duration ([17] and present studies), the presence of incontinence [13,19], symptom scores [13,15], and previous use of antimuscarinics [17]. Based on the overall evidence, we conclude that both greater baseline OAB severity and smaller improvements upon initial treatment tend to be associated with a decision for dose escalation. However, each of these associations appears to be too weak to be robustly detected across studies; this applies even more so if individual OAB symptoms are considered.

### 4.4. Factors Associated with Treatment Outcomes

While an understanding of allocation and escalation bias is interesting mechanistically, the clinically more relevant question is which factors are associated with treatment outcomes at study end. In the present studies, reductions in OAB symptoms were of comparable extent in the 30/30, 30/45 and 45/45 mg cohorts; if anything, the 45/45 cohort tended to have slightly greater and the 30/45 cohort slightly smaller efficacy as compared to the 30/30 group. For a more quantitative analysis, we have specifically looked at improvements of each of the four OAB parameters in separate logistic regression analyses. As previously reported from the NIS with other antimuscarinics [32], the strongest and most consistent contributor to improvement of a given symptom was the baseline value of that symptom. This reflects that the four symptoms are only moderately correlated to each other [31,39] and that a high baseline value for a given symptoms allows for a large reduction upon treatment. However, compared to the 45/45 group, starting and staying at 30 mg or escalating from 30 to 45 mg tended to have only minor and inconsistent effects achieving a *p*-value < 0.05 only for the 30/45 group for urgency and incontinence in study I but not for the other two outcome parameters within that study or for any outcome parameter in study II. While other studies have not applied such complex models to overall treatment outcomes, to the best of our knowledge, they have reported that patients with and without dose escalation achieved similar improvements of OAB symptoms at study end [13,14,18], although some reported slightly smaller improvements in escalators than in non-escalators [15].

## 5. Conclusions

Our analyses identified younger age as the only factor consistently associated with a greater starting dose, but the relative impact of age was small. Several drugs allow for a dose escalation if greater efficacy is desired and tolerability is adequate. While the overall evidence points to both greater baseline symptom intensity and smaller initial improvement, either effect is apparently too small to be detected consistently across studies, particularly when individual OAB parameters are considered. Most important, from a clinical perspective, is the observation in the present and in most previous studies that dose escalation in patients with an insufficient initial efficacy results in a symptom improvement comparable to that observed in patients exhibiting a greater initial improvement and staying on the lower dose or in those starting and staying on the higher dose. These data support the concept that dose escalation helps to achieve meaningful symptom improvements in patients where tolerability of the lower dose allows for a dose increase. In this regard, both dose strengths of propiverine had similar tolerability and discontinuations after the initial treatment period were similar with the starting doses of 30 and 45 mg.

## Figures and Tables

**Figure 1 jcm-10-00311-f001:**
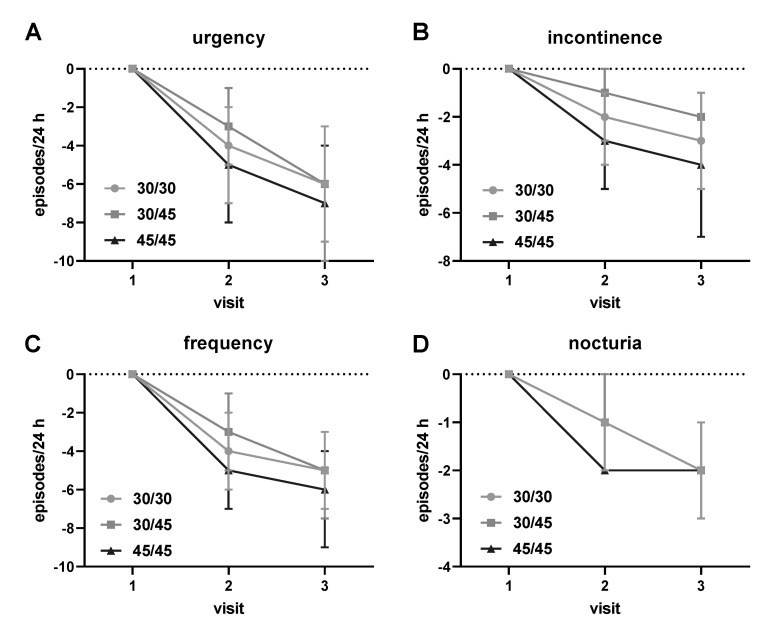
Intra-individual change of overactive bladder (OAB) symptoms of urgency (**A**), incon-tinence (**B**), frequency (**C**) and nocturia (**D**) in the cohorts of patients starting on 30 mg and staying on that dose (30/30), starting on 30 mg and escalating to 45 mg at visit 2 after about 4 weeks (30/45) and starting and staying on 45 mg until study end after about 12 weeks (45/45) in study I. Data are shown as medians with IQR. Means ± SD are shown for comparison in Appendix A. Patients not exhibiting a given symptom at baseline were excluded from the analysis of that symptom; specifically, 364 subjects reported no incontinence at baseline. Corresponding data for study II are shown in the Appendix A.

**Table 1 jcm-10-00311-t001:** Demographic and OAB-related baseline variables in patients starting treatment with a propiverine dose of 30 or 45 mg/d.

	Initial 30 mg	Initial 45 mg	*p*-Value
*n*	1021	239	
**Demographic parameters**			
Gender, % female/male	64.3/32.6	64.3/32.9	
Previous OAB treatment, %	33.7	45.0	
Age, years	65.8 ± 13.1	65.4 ± 12.5	0.3820
Height, cm	169.0 ± 7.9	169.9 ± 8.3	0.1543
Weight, kg	77.4 ± 14.2	80.3 ± 17.0	0.0184
BMI, kg/m^2^	27.1 ± 4.3	27.9 ± 5.5	0.1187
**OAB-related parameters**			
OAB duration, months	12.1 (3.9; 34.6)	13.2 (4.7; 35.1)	0.3343
Urgency episodes/24 h	9 (6; 13) 10.0 ± 5.8	10 (7; 14) 10.8 ± 5.8	0.0251
Incontinence episodes/24 h	6 (4; 23) 4.7 ± 3.7	8 (5; 30) 6.2 ± 4.4	<0.0001
Urinary frequency/24 h	13 (11; 16) 13.3 ± 4.2	14 (11; 17) 14.3 ± 4.4	0.0004
Nocturia episodes/24 h	3 (2; 4) 3.4 ± 1.6	3 (3; 5) 3.7 ± 1.7	0.0008

Data are shown as % of patients for gender (does not add up to 100 due to missing values), as means ± SD for continuous demographic parameters, and medians with inter-quartile range (IQR) of OAB-related parameters (means ± SD only shown to facilitate comparison with previous reports). Descriptive *p*-values for the difference between groups are from univariate analysis using unpaired, two-tailed Kruskal–Wallis tests. The analysis of the OAB symptoms included only patients that had a documented dose and a measured value at baseline other than 0.

**Table 2 jcm-10-00311-t002:** Factors associated with starting dose (30 vs. 45 mg) in a logistic regression analysis of studies I and II. Data are the reported parameter for maximum likelihood estimate with its standard error (SE) and descriptive *p*-value based on Chi square tests. Parameters with a *p* > 0.3 were not retained in the model and are not shown.

Parameter	Estimate ± SE	*p*-Value	Estimate ± SE	*p*-Value
	Study I		Study II	
Gender, female	−0.225 ± 0.140	0.1081	-	-
Age, years	0.017 ± 0.008	0.0335	0.027 ± 0.010	0.0067
Weight, kg	0.083 ± 0.070	0.2338	−0.132 ± 0.093	0.1556
Height, cm	−0.120 ± 0.069	0.0820	0.010 ± 0.090	0.2706
BMI, kg/m^2^	−0.261 ± 0.198	0.1875	0.347 ± 0.271	0.2013
OAB duration, months	-	-	−0.011 ± 0.002	<0.0001
Urgency/24 h	-	-	0.032 ± 0.023	0.1606
Incontinence/24 h	−0.107 ± 0.024	<0.0001	-	-
Micturitions/24 h	-	-	−0.238 ± 0.071	0.0008
Nocturia/24 h	−0.126 ± 0.060	0.0348	-	-

**Table 3 jcm-10-00311-t003:** Demographic and OAB-related baseline variables and after 4 weeks in patients starting treatment with a propiverine dose of 30 mg/d and either staying on that dose after 4 weeks or increasing it 45 mg/d.

	Stay on 30 mg	Increase to 45 mg	*p*-Value
*n*	789	160	
Demographic parameters			
Gender, % female/male	66.2/33.8	60.0/40.0	
Previous OAB treatment, %	31.1	50.6	
Age, years	65.6 ±13.1	67.0 ± 13.0	0.2313
Height, cm	169.0 ± 7.7	171.3 ± 7.9	0.0010
Weight, kg	77.0 ± 13.9	81.2 ± 14.4	<0.0001
BMI, kg/m^2^	27.0 ± 4.2	27.6 ± 4.6	0.1403
OAB-related parameters at baseline			
OAB duration, months	11.2 (3.6; 31.3)	13.6 (5.0; 51.9)	
Urgency episodes/24 h	9 (5; 13) 9.6 ± 5.6	11 (6.5; 15) 11.7 ± 6.6	<0.0001
Incontinence episodes/24 h	4 (2; 6) 4.6 ± 3.4	4 (2; 7) 5.4 ± 4.7	<0.0001
Urinary frequency/24 h	13 (10; 15) 13.1 ± 4.1	14 (12; 16) 14.3 ± 4.1	<0.0001
Nocturia/24 h	3 (2; 4) 3.3 ± 1.5	4 (3; 4.5) 3.7 ± 1.6	<0.0001
OAB-related parameters after 4 weeks			
Urgency episodes/24 h	3 (2; 6) 4.4 ± 3.6	7 (4; 11) 8.0± 5.4	<0.0001
Incontinence episodes/24 h	1 (0; 2) 1.8 ± 2.1	2 (1; 5) 3.6 ± 3.6	<0.0001
Urinary frequency/24 h	8 (7; 10) 8.8 ± 2.8	11 (9; 13) 11.2 ± 3.4	<0.0001
Nocturia/24 h	2 (1; 2) 1.8 ± 1.1	2 (2; 3) 2.7 ± 1.2	<0.0001

Data are shown as % of patients for gender, as means ± SD for continuous demographic parameters, and medians with IQR of OAB-related parameters (means ± SD only shown to facilitate comparison with previous reports). Descriptive *p*-values for the difference between groups are from univariate analysis using unpaired, two-tailed Kruskal–Wallis tests. The analysis of the OAB symptoms included only patients that had a documented dose and a measured value at baseline other than 0.

**Table 4 jcm-10-00311-t004:** Factors associated with staying at the starting dose of 30 mg vs. increasing to a dose of 45 mg in a logistic regression analysis taking both OAB parameters at baseline and after 4 weeks into consideration.

Parameter	Estimate ± SE	*p-*value	Estimate ± SE	*p*-Value
	Study I		Study II	
Age, years	-	-	1.420 ± 1.387	0.3059
Weight, kg	−0.017 ± 0.009	0.0611	−0.17 ± 0.012	0.1444
Height, cm	−0.036 ± 0.018	0.0413	-	-
Urgency/24 h baseline	−0.100 ± 0.036	0.0049	-0.192 ± 0.054	0.0004
Incontinence/24 h baseline	−0.087 ± 0.053	0.0998	0.108 ± 0.078	0.1659
Micturitions/24 h baseline	−0.069 ± 0.056	0.2185	-	-
Nocturia/24 h baseline	−0.160 ± 0.119	0.1776	−0.311 ± 0.155	0.0450
Urgency/24 h 4 weeks	−0.067 ± 0.037	0.0669	-	-
Incontinence/24 h 4 weeks	−0.126 ± 0.058	0.0300	−0.201 ±00.084	0.0165
Micturitions/24 4 weeks	-	-	-	-
Nocturia/24 h 4 weeks	-	-	−0.549 ± 0.227	0.0153

Data are reported parameter estimate for maximum likelihood estimate with its standard error (SE) and descriptive *p*-value based on Chi square tests. Parameters with a *p* > 0.3 were not retained in the model and are not shown.

**Table 5 jcm-10-00311-t005:** Factors associated with overall improvement of urgency (12 weeks vs. baseline) in a logistic regression analysis taking demographics, OAB parameters at baseline, duration of condition and dose level into consideration.

Parameter	Estimate ± SE	*p*-Value	Estimate ± SE	*p*-Value
	Study I		Study II	
Gender, female	0.026 ± 0.009	0.0056	0.011 ± 0.414	0.3954
Age, years	0.026 ± 0.009	0.0056	0.011 ± 0.012	0.3739
Weight, kg	−0.123 ± 0.081	0.1592	0.019 ± 0.112	0.8628
Height, cm	0.114 ± 0.081	0.1592	−0.028 ± 0.109	0.7973
BMI, kg/m^2^	0.340 ± 0.235	0.1475	−0.039 ± 0.327	0.9049
OAB duration, months	0.012 ± 0.002	<0.0001	0.005 ± 0.003	0.0677
Urgency/24 h	−0.714 ± 0.025	<0.0001	−0.842 ± 0.034	<0.0001
Incontinence/24 h	−0.003 ± 0.035	0.9282	0.210 ± 0.043	<0.0001
Micturitions/24 h	−0.008 ± 0.037	0.8364	0.029 ± 0.054	0.5928
Nocturia/24 h	−0.043 ± 0.008	0.6264	−0.087 ± 0.115	0.4483
Dose 30/30 *	0.109 ± 0.297	0.7138	−0.428 ± 0.328	0.1917
Dose 30/45 *	0.998 ± 0.373	0.0076	0.343± 0.490	0.4483

*p*-values for gender relate to male and those for dose level relate to the 45/45 group as reference; *: term not uniquely estimable.

**Table 6 jcm-10-00311-t006:** Factors associated with overall improvement of incontinence (12 weeks vs. baseline) in a logistic regression analysis taking demographics, OAB parameters at baseline, duration of condition and dose level into consideration.

Parameter	Estimate ± SE	*p*-Value	Estimate ± SE	*p*-Value
	Study I		Study II	
Gender, female	−0.012 ± 0.178	0.2662	−0.331 ± 0.261	0.2060
Age, years	0.012 ± 0.005	0.0182	0.003 ± 0.008	0.7324
Weight, kg	−0.048 ± 0.046	0.3025	−0.051 ± 0.071	0.4701
Height, cm	0.042 ± 0.046	0.3741	0.043 ± 0.069	0.5347
BMI, kg/m^2^	0.130 ± 0.132	0.3257	0.156 ± 0.206	0.4498
OAB duration, months	0.008 ± 0.001	<0.0001	0.003 ± 0.002	0.1493
Urgency/24 h	−0.008 ± 0.014	0.5685	−0.056 ± 0.021	0.0086
Incontinence/24 h	−0.765 ± 0.020	<0.0001	−0.657 ± 0.028	<0.0001
Micturitions/24 h	−0.018 ± 0.021	0.3851	0.011 ± 0.035	0.7559
Nocturia/24 h	−0.032 ± 0.050	0.3851	0.022 ± 0.072	0.7655
Dose 30/30 *	−0.103 ± 0.168	0.5384	−0.247 ± 0.208	0.2359
Dose 30/45 *	0.487 ± 0.211	0.0211	−0.128 ± 0.306	0.6758

*p*-values for gender relate to male and those for dose level relate to the 45/45 group as reference; *: term not uniquely estimable.

## Data Availability

Data are owned by Apogepha (Dresden, Germany). They will be made available to qualified non-commercial investigators upon request to the corresponding author.

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
