# Peer review of "Factors Associated with Decisions for Initial Dosing, Up-Titration of Propiverine and Treatment Outcomes in Overactive Bladder Syndrome Patients in a Non-Interventional Setting"

_jcm, 2021, doi:10.3390/jcm10020311_

Round 1

Reviewer 1 Report

This is an excellent study.

The objectives are clear and the methods are sound and the conclusions are consistent with the mission. 

The authors elucidate the difference between a NIS and a randomized study and the implications.

Given the number of antimuscarinic agents and the "double-dose" option, it is interesting that the choice of 30/45 was elected in this study.   

The NIS was consistent with real-life prescribing - and the only option not provided was lowering the dose if 45mg (as a starting dose) was not tolerated. In fact, as a clinician, I might always start with 45mg based on this data and reserve 30mg for patients who are "sensitive" to an AE.

With the possibility of understanding allocation dosing bias, it would appear that one would have to survey the prescribers! Why did you do that ...? This is a question for the doctors, not the demographics.

Also, understood is that the cohort becomes biased - this is not a dose-escalation study if patients start on the higher dose - and it is unknown how many of the patients would NOT have escalated or in this group if they would have de-escalated.  This is not a flaw of the study, just an observation of the possibility matrix. 

This is an excellent contribution to the literature.

Author Response

This is an excellent study.

The objectives are clear and the methods are sound and the conclusions are consistent with the mission. 

The authors elucidate the difference between a NIS and a randomized study and the implications.

Given the number of antimuscarinic agents and the "double-dose" option, it is interesting that the choice of 30/45 was elected in this study.   

The NIS was consistent with real-life prescribing - and the only option not provided was lowering the dose if 45mg (as a starting dose) was not tolerated. In fact, as a clinician, I might always start with 45mg based on this data and reserve 30mg for patients who are "sensitive" to an AE.

With the possibility of understanding allocation dosing bias, it would appear that one would have to survey the prescribers! Why did you do that ...? This is a question for the doctors, not the demographics.

Also, understood is that the cohort becomes biased - this is not a dose-escalation study if patients start on the higher dose - and it is unknown how many of the patients would NOT have escalated or in this group if they would have de-escalated.  This is not a flaw of the study, just an observation of the possibility matrix. 

This is an excellent contribution to the literature.

Reply: We highly appreciate that the reviewer feels so positive about our manuscript. We understood these comments to mean that there was no request for specific changes but an opportunity to reply to some of the comments:

  • We as authors cannot comment on why the company holding the marketing authorization has chosen to apply for registration of the 30 mg and 45 mg doses and not, for instance, a 60 mg dose. However, these are the two approved doses in Germany. The original dose-response studies in patients with OAB and neurogenic detrusor overactivity have shown no greater efficacy at doses > 45 mg, but more side effects/adverse events. (DOI 3109/00365599509180578 for OAB; Mazur et al. Urologe [A] 1994; 33:447-452 for neurogenic)
  • There may be a misunderstanding: Starting at 45 mg and lowering after about 4 weeks was an option. However, this was chosen in too few cases to merit analysis of this group (28 and 10 patients in studies I and II, see l. 207 in main manuscript and supplementary Figures 1 and 2).
  • With hindsight, we agree that such questions should have been posed to the prescribing physicians. However, we feel that there is merit in looking at objective parameters associated with the decisions. For this reason, we consistently talk about “factors associated” and not “reasons” for dosing decisions.

Reviewer 2 Report

Thank you for this interesting and thorough study. It’s a little difficult for the reader due to the multiple studies and research questions. Mostly, I think the reader would benefit from an extended table 3 in the main paper (and maybe remove some of the listed results in the text into a table). If necessary leave mean(SD) in the supplement for space/completeness.

If the methodology of the two studies was so very similar, I wonder if it would be possible to combine the two (separate analysis could be put in the supplement?) This would make this paper so much more accessible to the reader. If this is not the case, you should explain the factors that make the two studies different and how they might affect the differing results. For instance, did study I and II use the same physicians? Was there a geographical/system/provider difference between the two studies which might cause differences in prescribing techniques? Was there a difference in missing data/quality of reported data between the two? Different inclusion criteria?

Did all participants have urgency urinary incontinence? If not was ‘0’ baseline incontinence episodes treated as a continuous variable or were those patients excluded from the analysis of that variable?

This was confusing notation: ‘a reduction in urgency episodes by 4 (2; 7; 5.1± 4.6)’. Assuming the median is outside the parentheses and 2 is the lower quartile and 7 the UQ, please either write the IQR as a single number or a hyphenated range like 4(5; 5.1± 4.6) or 4(2-7; 5.1± 4.6).

Please include changes in OAB parameters in table 3.

In 3.4, you state that greater improvement is associated with dose escalation, and that doesn’t seem to be the case – the smallest changes are in the escalation group.

I think that the reader would benefit from some of the data tables in the supplement being in the actual paper. In fact, I think it would be much more helpful for the reader to have all OAB results in one place – it is very difficult to try to read some in the text and then compare to a table in the supplement.

Since OAB parameters were associated with dose choice and escalation, baseline parameters are different between the groups. Might it therefore be helpful to consider a percentage improvement? Additionally, might it be worth considering the absolute value at week 4 as a predictor, assuming a physician might evaluate this against ‘tolerable’ ranges.

There is no table 6 in the supplementary material.

What do you mean by people who discontinued treatment ‘after the treatment period’? Were these people who completed the treatment but were lost to follow-up? It seems from the next sentence that all 145 discontinued during the treatment period. What was the breakdown of those who discontinued after visit 2 who had the escalated dose?

You state you have comedications but do not examine factors (e.g. anticholinergic burden) as a decision for starting at a lower dose.  

There was no mention of the comparison of ADRs in each of the three dose regimens in study II.

You state height as a variable associated with dose escalation, but since you did not test sex (I think you should be reporting sex, not gender, here) might height be a function of whether men are more likely to have dose escalation?

A few typing errors:

Is this missing the end of the sentence: ‘Post-hoc analysis of data from two non-interventional studies of 1335 and 745 OAB patients, respectively, receiving treatment with propiverine.’

Missing no. of weeks []: ‘Patients starting on the 30 mg dose and escalating to 45 mg after [] weeks had outcomes comparable with those staying on a starting dose of 30 or 45 mg.’

Confusing word order: Rather participating 456 and 158 physicians… Consider: Rather, 456 and 158 participating physicians… 

Missing [the]? ‘The model for treatment outcomes also included the same variables and addition-ally [the] dose after visits 1 and 2.’

Author Response

General reply: We must apologize. Our submission was intended to include 6 tables in the main manuscript. While all were mentioned in the main text, they apparently got “lost” in the final uploaded version. We have now reintroduced them and hope that this helps to address some of the specific questions.

Thank you for this interesting and thorough study. It’s a little difficult for the reader due to the multiple studies and research questions. Mostly, I think the reader would benefit from an extended table 3 in the main paper (and maybe remove some of the listed results in the text into a table). If necessary leave mean(SD) in the supplement for space/completeness.

Reply: We are happy to learn that this reviewer found the study to be interesting and thorough. During the writing process we have indeed struggled to find a good balance between full and transparent reporting vs. maintaining readability. This was a challenging task given the multitude of different populations (full study group, 30/30, 30/45 and 45/45) and each of them having specific baseline parameters. As a result, we had already generated an Online Supplement of 13 single-spaced pages, but apparently this still left readers with a rather complex main document. Specifically, we have considered removing means ± SD to the Online Supplement as well. However, we decided against this for two reasons: Firstly, it would lead to duplication of all applicable tables, thereby potentially causing confusion. Second, the reporting of means ± SD is so prevalent in the published literature (see Amiri 2020), that we felt that providing such data for comparison in the main text provides more benefit than it creates problems with readability.

If the methodology of the two studies was so very similar, I wonder if it would be possible to combine the two (separate analysis could be put in the supplement?) This would make this paper so much more accessible to the reader. If this is not the case, you should explain the factors that make the two studies different and how they might affect the differing results. For instance, did study I and II use the same physicians? Was there a geographical/system/provider difference between the two studies which might cause differences in prescribing techniques? Was there a difference in missing data/quality of reported data between the two? Different inclusion criteria?

Reply: While the two studies were rather similar in design, they had distinct primary aims and were performed at different times. One of us (MCM) regularly teaches statistics courses and has been a regular statistical reviewer for J Urol. Based on this bona fide expertise, we wish to explain that a pooled analysis of two similar but distinct datasets is not the same as reporting both of them in parallel. While this creates added complexity, we consider it a key message of the manuscript that even under such similar conditions, two studies can yield different answers to the same question. We find this very helpful in understanding why previously reported studies for the very same questions (specific factors associated with dosing decisions) also reached heterogeneous answers. We had also discussed internally the option to report each study in a separate manuscript. However, given that the two did not reach the same conclusions on some questions, we felt that this would not only look as salami tactics but also generate impressions in one paper that were not confirmed in the other. Therefore, we would like to stick to our decision to report both studies separately.

We are not sure what the referee means by “prescribing techniques”. While prescribing habits may have shifted over time, we have no evidence for differences in geographic representation. System/provider differences are not applicable to the German healthcare system. Moreover, we have also no evidence for a systematic difference in missing values between the studies. Finally, neither study had inclusion/exclusion criteria other than the Summary of Product Characteristics, which did not change between studies I and II.

Did all participants have urgency urinary incontinence? If not was ‘0’ baseline incontinence episodes treated as a continuous variable or were those patients excluded from the analysis of that variable?

Reply: We apologize for not having specifically reported the percentage of patients with dry vs. wet OAB. We have now added those numbers to the manuscript; shortly, 364/1149 patients with reported incontinence data in study I 171/589 had OAB dry (l. 187-188 and legends to Supplementary Figures 3-5). As explained in the manuscript (l. 150-152), patients not exhibiting a given symptom at baseline were not included in the analysis of changes of that symptom over time.

This was confusing notation: ‘a reduction in urgency episodes by 4 (2; 7; 5.1± 4.6)’. Assuming the median is outside the parentheses and 2 is the lower quartile and 7 the UQ, please either write the IQR as a single number or a hyphenated range like 4(5; 5.1± 4.6) or 4(2-7; 5.1± 4.6).

Reply: We have adopted the suggestion and now consistently report IQR as ranges (e.g. 2-7). However, we would like to alert the reviewer to the fact that this leads to the strange writing of e.g. -5—1 in cases where the lower end is -5 and the upper end -1.

Please include changes in OAB parameters in table 3.

Reply: The way our model was run, it included not the change at week 4 directly but rather measured values at baseline and at week 4. Therefore, reporting those values (and not the change) is the appropriate way of showing these data in Table 3. However, changes in these parameters are shown in Figure 1 and Supplementary Figure 3-5.

In 3.4, you state that greater improvement is associated with dose escalation, and that doesn’t seem to be the case – the smallest changes are in the escalation group.

Reply: There may be a misunderstanding: the table does not show changes but rather measured values at baseline and week 4. The legend of this table was revised for clarity, but the statements in the main text are correct.

I think that the reader would benefit from some of the data tables in the supplement being in the actual paper. In fact, I think it would be much more helpful for the reader to have all OAB results in one place – it is very difficult to try to read some in the text and then compare to a table in the supplement.

Reply: We are open to this suggestion but would like to raise the issue that the referee proposed above the opposite, i.e. shifting more data from the main manuscript to the Online Supplement. The suggestion may have come from the fact that the tables intended for the main manuscript apparently got lost in the finally formatting steps. If the referee continues to recommend shifting information from the Online Supplement to the main manuscript after having seen the lost tables, we will be happy to implement that suggestion.

Since OAB parameters were associated with dose choice and escalation, baseline parameters are different between the groups. Might it therefore be helpful to consider a percentage improvement? Additionally, might it be worth considering the absolute value at week 4 as a predictor, assuming a physician might evaluate this against ‘tolerable’ ranges.

Reply: There are two ways the models can be set up to perform the analyses we intended. One way is to include absolute values at baseline and week 4, the other is to consider % changes. We had chosen the former approach because % changes already include baseline values in their calculations; as we also wanted to account for allocation bias, we had to include baseline values in the model. Had we included baseline and % changes, baseline would indirectly have entered the model twice, which would have contaminated the outcomes. Considering the absolute value at week 4 is what we have done (l. 153-154).

There is no table 6 in the supplementary material.

Reply: This may be a misunderstanding. All tables mentioned in the main manuscript are part of it, but due to a technical glitch had disappeared from the submitted version. They have now been re-added.

What do you mean by people who discontinued treatment ‘after the treatment period’? Were these people who completed the treatment but were lost to follow-up? It seems from the next sentence that all 145 discontinued during the treatment period. What was the breakdown of those who discontinued after visit 2 who had the escalated dose?

Reply: As our manuscript represents a post-hoc analysis of an existing database, the authors had no influence on how data were captured. While an observation period of about 12 weeks had been specified, the pre-specified statistical analysis plan counted patients stopping treatment after this as adverse drug reaction. This certainly is open to debate. However, it reflects that this was an observational study, i.e. nobody was treated for study purposes and only treatments initiated by the physician based on his decision were documented systematically; thus, stopping treatment after 12 weeks can be seen as adverse drug reaction. In any case, we as authors did not wish to down-size reported adverse drug reactions by changing how they were counted; most importantly, as an employee of the sponsoring company is an author, we want to avoid any suspicion that data on tolerability had been “manipulated”.

You state you have comedications but do not examine factors (e.g. anticholinergic burden) as a decision for starting at a lower dose.

Reply: This is an excellent idea that we had not considered before. However, given that the editor gave us only 6 days to resubmit, and 4 of those are public holidays or weekend days in Germany, we cannot do within the allotted time frame, particularly as any calculations would have to be done by the contract research organization based in Switzerland. Moreover, introduction of any new variable would require to redo all calculations in the manuscript and would certainly affect more or less any reported number to some extent.

There was no mention of the comparison of ADRs in each of the three dose regimens in study II.

Reply: This is correct. We deemed absolute numbers within each group to be too small to make this meaningful.

You state height as a variable associated with dose escalation, but since you did not test sex (I think you should be reporting sex, not gender, here) might height be a function of whether men are more likely to have dose escalation?

Reply: There may be a misunderstanding. Gender was included in all models; it is not reported in some outcome tables because it did not fulfill the criteria for retention in the model in some cases. Thus, parameter estimates for gender are included in Tables 2, 5 and 6.

A few typing errors:

Is this missing the end of the sentence: ‘Post-hoc analysis of data from two non-interventional studies of 1335 and 745 OAB patients, respectively, receiving treatment with propiverine.’

Missing no. of weeks []: ‘Patients starting on the 30 mg dose and escalating to 45 mg after [] weeks had outcomes comparable with those staying on a starting dose of 30 or 45 mg.’

Confusing word order: Rather participating 456 and 158 physicians… Consider: Rather, 456 and 158 participating physicians… 

Missing [the]? ‘The model for treatment outcomes also included the same variables and addition-ally [the] dose after visits 1 and 2.’

Reply: All typing errors have been fixed. Thanks for spotting them!

Round 2

Reviewer 2 Report

Thank you for the fast response. I have tried to assess the changes you made to the manuscript but this journal does not appear to provide a ‘track change’ version. I therefore apologise in advance if I miss anything you addressed in the main text. I appreciate your replies to my comments, please see my follow-up comments interspersed with the dialogue denoted by **

Thank you for this interesting and thorough study. It’s a little difficult for the reader due to the multiple studies and research questions. Mostly, I think the reader would benefit from an extended table 3 in the main paper (and maybe remove some of the listed results in the text into a table). If necessary leave mean(SD) in the supplement for space/completeness.

Reply: We are happy to learn that this reviewer found the study to be interesting and thorough. During the writing process we have indeed struggled to find a good balance between full and transparent reporting vs. maintaining readability. This was a challenging task given the multitude of different populations (full study group, 30/30, 30/45 and 45/45) and each of them having specific baseline parameters. As a result, we had already generated an Online Supplement of 13 single-spaced pages, but apparently this still left readers with a rather complex main document. Specifically, we have considered removing means ± SD to the Online Supplement as well. However, we decided against this for two reasons: Firstly, it would lead to duplication of all applicable tables, thereby potentially causing confusion. Second, the reporting of means ± SD is so prevalent in the published literature (see Amiri 2020), that we felt that providing such data for comparison in the main text provides more benefit than it creates problems with readability.

**I completely understand the issue. As a reader I found myself going back and forth to the supplement to reference tables 2 and 3, hence I thought they might be useful in the main paper. My comment re: moving the means/SD to the supplement were purely to address any space or clutter concerns you or the journal may have.  

If the methodology of the two studies was so very similar, I wonder if it would be possible to combine the two (separate analysis could be put in the supplement?) This would make this paper so much more accessible to the reader. If this is not the case, you should explain the factors that make the two studies different and how they might affect the differing results. For instance, did study I and II use the same physicians? Was there a geographical/system/provider difference between the two studies which might cause differences in prescribing techniques? Was there a difference in missing data/quality of reported data between the two? Different inclusion criteria?

Reply: While the two studies were rather similar in design, they had distinct primary aims and were performed at different times. One of us (MCM) regularly teaches statistics courses and has been a regular statistical reviewer for J Urol. Based on this bona fide expertise, we wish to explain that a pooled analysis of two similar but distinct datasets is not the same as reporting both of them in parallel. While this creates added complexity, we consider it a key message of the manuscript that even under such similar conditions, two studies can yield different answers to the same question. We find this very helpful in understanding why previously reported studies for the very same questions (specific factors associated with dosing decisions) also reached heterogeneous answers. We had also discussed internally the option to report each study in a separate manuscript. However, given that the two did not reach the same conclusions on some questions, we felt that this would not only look as salami tactics but also generate impressions in one paper that were not confirmed in the other. Therefore, we would like to stick to our decision to report both studies separately.

** I understand your rationale for this (I was considering perhaps a meta analysis rather than pooled data, although this might also be impractical with only two studies). I think it’s very important to note how very similar the studies were in design and population, and yet the differing outcomes, and I think it has important implications in comparison with other similar studies. I think it is worth addressing the decision to present both separately as a strength and a cautionary note to be circumspect when interpreting such studies alone.

We are not sure what the referee means by “prescribing techniques”. While prescribing habits may have shifted over time, we have no evidence for differences in geographic representation. System/provider differences are not applicable to the German healthcare system. Moreover, we have also no evidence for a systematic difference in missing values between the studies. Finally, neither study had inclusion/exclusion criteria other than the Summary of Product Characteristics, which did not change between studies I and II.

**Thank you. I was particularly probing for potential systematic differences between the two studies which might explain outcome differences, related to my previous point. I think the reader might benefit from being explicitly told that you have not identified any differences which may affect study outcomes, and that prescribing recommendations/habits are reasonably homogeneous across the sample (e.g. per national guidelines).  

This was confusing notation: ‘a reduction in urgency episodes by 4 (2; 7; 5.1± 4.6)’. Assuming the median is outside the parentheses and 2 is the lower quartile and 7 the UQ, please either write the IQR as a single number or a hyphenated range like 4(5; 5.1± 4.6) or 4(2-7; 5.1± 4.6).

Reply: We have adopted the suggestion and now consistently report IQR as ranges (e.g. 2-7). However, we would like to alert the reviewer to the fact that this leads to the strange writing of e.g. -5—1 in cases where the lower end is -5 and the upper end -1.

** If the notation is confusing, your original notation could be used but please state that you provide the lower quartile; upper quartile rather than IQR to avoid confusion about the extra number.

In 3.4, you state that greater improvement is associated with dose escalation, and that doesn’t seem to be the case – the smallest changes are in the escalation group.

Reply: There may be a misunderstanding: the table does not show changes but rather measured values at baseline and week 4. The legend of this table was revised for clarity, but the statements in the main text are correct.

**There is still a misunderstanding! You state in 3.4: “A greater number of urgency episodes at baseline and a greater improvement of incon-tinence episodes after 4 weeks of treatment were associated with dose escalation in both studies with a p < 0.05;” In study 1, the dose escalation group had 11 vs 9 urgency episodes in the non-escalation group, therefore greater urgency was associated with dose escalation. However, greater improvement in incontinence implies a greater reduction in incontinence episodes. Improvement in incontinence is not included in the table (this is why I suggested including change values in the tables, since you reference change values here). Looking at the graphs in figure 1, the dose escalation group had a reduction of 1 incontinence episode after 4 weeks, and the 30/30 group had a reduction of 2 episodes, which would suggest that dose escalation is associated with a smaller improvement in incontinence. In fact, in the discussion, you state “Adding to this heterogeneity in findings, dose-escalation was associated with greater baseline values and/or smaller initial improvements sometimes for all OAB symptoms, but sometimes such as also observed in the present study for only selected OAB symptoms.” Which, if I interpret this sentence correctly, seems to suggest that dose escalation in this study is associated with smaller initial improvements in some OAB symptoms. You also conclude that “Based on the overall evidence we conclude that both greater baseline OAB severity and smaller improvements upon initial treatment tend to be associated with a decision for dose escalation.” Both statements contradict the result you describe in 3.4. If the statement that greater improvements in incontinence is associated with dose escalation is indeed correct per the logistic regression model (although table 4 does not seem to suggest that), I think that this should be examined and discussed closely in the paper as it is counter-intuitive, both from the data provided and in practice. (This is also applicable to the improvement in number of nocturia episodes.)

I think that the reader would benefit from some of the data tables in the supplement being in the actual paper. In fact, I think it would be much more helpful for the reader to have all OAB results in one place – it is very difficult to try to read some in the text and then compare to a table in the supplement.

Reply: We are open to this suggestion but would like to raise the issue that the referee proposed above the opposite, i.e. shifting more data from the main manuscript to the Online Supplement. The suggestion may have come from the fact that the tables intended for the main manuscript apparently got lost in the finally formatting steps. If the referee continues to recommend shifting information from the Online Supplement to the main manuscript after having seen the lost tables, we will be happy to implement that suggestion.

**I hope I was clearer in my intentions above – my suggestion of movement of means/SD to the supplement was a compromise if there was a space concern adding tables to the main paper. I certainly think the demographic table and current table 3 are useful additions to the main paper (I would also advocate for the corresponding study II table to be included). I am of the opinion that data tables with common summary statistics like this are of the most use to the reader and help the reader understand the paper. For the same reason, I appreciate your inclusion of means/SD. I don’t necessarily think that the tables of parameter estimates from the regressions need to be in the main paper if space is a concern, since these are less intuitive to the average clinical reader.

Since OAB parameters were associated with dose choice and escalation, baseline parameters are different between the groups. Might it therefore be helpful to consider a percentage improvement? Additionally, might it be worth considering the absolute value at week 4 as a predictor, assuming a physician might evaluate this against ‘tolerable’ ranges.

Reply: There are two ways the models can be set up to perform the analyses we intended. One way is to include absolute values at baseline and week 4, the other is to consider % changes. We had chosen the former approach because % changes already include baseline values in their calculations; as we also wanted to account for allocation bias, we had to include baseline values in the model. Had we included baseline and % changes, baseline would indirectly have entered the model twice, which would have contaminated the outcomes. Considering the absolute value at week 4 is what we have done (l. 153-154).

**The legend for table 4 suggests that “taking both OAB parameters after 4 and their change as compared to baseline into consideration” (I assume 4 weeks?), but the preceding results talks about urgency at baseline being associated. I had assumed that symptoms/24h in the table are baseline and ‘changes’ are taking both time points into account, but I think this needs a better explanation, since the legend suggests these are at the 4 week time point, in which case baseline values are not shown.

What do you mean by people who discontinued treatment ‘after the treatment period’? Were these people who completed the treatment but were lost to follow-up? It seems from the next sentence that all 145 discontinued during the treatment period. What was the breakdown of those who discontinued after visit 2 who had the escalated dose?

Reply: As our manuscript represents a post-hoc analysis of an existing database, the authors had no influence on how data were captured. While an observation period of about 12 weeks had been specified, the pre-specified statistical analysis plan counted patients stopping treatment after this as adverse drug reaction. This certainly is open to debate. However, it reflects that this was an observational study, i.e. nobody was treated for study purposes and only treatments initiated by the physician based on his decision were documented systematically; thus, stopping treatment after 12 weeks can be seen as adverse drug reaction. In any case, we as authors did not wish to down-size reported adverse drug reactions by changing how they were counted; most importantly, as an employee of the sponsoring company is an author, we want to avoid any suspicion that data on tolerability had been “manipulated”.

**I *think* I understand this. Please make this more clear in the text “those that discontinued drug upon conclusion of the observation period of this study are included in these numbers etc…”. The current sentence is still confusing.

You state height as a variable associated with dose escalation, but since you did not test sex (I think you should be reporting sex, not gender, here) might height be a function of whether men are more likely to have dose escalation?

Reply: There may be a misunderstanding. Gender was included in all models; it is not reported in some outcome tables because it did not fulfill the criteria for retention in the model in some cases. Thus, parameter estimates for gender are included in Tables 2, 5 and 6.

** Thank you for clarifying. Importantly, are you actually reporting gender (social/cultural roles/personal identity; man/woman/other) rather than sex (biological attributes; male/female)? Please confirm or switch ‘gender’ to ‘sex’.

Author Response

Reviewer’s comment: Thank you for the fast response. I have tried to assess the changes you made to the manuscript but this journal does not appear to provide a ‘track change’ version. I therefore apologise in advance if I miss anything you addressed in the main text. I appreciate your replies to my comments, please see my follow-up comments interspersed with the dialogue denoted by **

Reply: We agree, a track-changes version is easier to read, and we had prepared one for the internal discussions among the authors. Unfortunately, there was no way to share this with the reviewer. Therefore, we had tried to facilitate identifying changes by specifically quoting the lines in which the changes were made. We also apply this to the present replies to your comments. All specific replies were identified as such following your ** paragraphs and identified as “Reply 2”. Another general note: As the journal has no length limits and the editor did not ask for any shortening, we have not implemented any suggestion aimed at shortening, unless we felt that this would make reading easier by removing unnecessary complexity.

Reviewer’s comment: Thank you for this interesting and thorough study. It’s a little difficult for the reader due to the multiple studies and research questions. Mostly, I think the reader would benefit from an extended table 3 in the main paper (and maybe remove some of the listed results in the text into a table). If necessary leave mean(SD) in the supplement for space/completeness.

Reply: We are happy to learn that this reviewer found the study to be interesting and thorough. During the writing process we have indeed struggled to find a good balance between full and transparent reporting vs. maintaining readability. This was a challenging task given the multitude of different populations (full study group, propiverine 30/30, 30/45 and 45/45) and each of them having specific baseline parameters. As a result, we had already generated an Online Supplement of 13 single-spaced pages, but apparently this still left readers with a rather complex main document. Specifically, we have considered removing means ± SD in the Online Supplement as well. However, we decided against this for two reasons: Firstly, it would lead to duplication of all applicable tables, thereby potentially causing confusion. Second, the reporting of means ± SD is so prevalent in the published literature (see Amiri 2020), that we felt that providing such data for comparison in the main text provides more benefit than it creates problems with readability.

Reviewer’s comment: **I completely understand the issue. As a reader I found myself going back and forth to the supplement to reference tables 2 and 3, hence I thought they might be useful in the main paper. My comment re: moving the means/SD to the supplement were purely to address any space or clutter concerns you or the journal may have.

Reply 2: Thank you for this suggestion. Supplementary Tables 2 and 3 are identical to main Tables 1 and 3, except that the former show data from study II whereas the latter show data from study I. While the Supplementary Tables 2 and 3 are necessary for diligent reporting of the data from the second study, they do not add to the story. Therefore, we prefer to keep them in the Online Supplement.

Reviewer’s comment: If the methodology of the two studies was so very similar, I wonder if it would be possible to combine the two (separate analysis could be put in the supplement?) This would make this paper so much more accessible to the reader. If this is not the case, you should explain the factors that make the two studies different and how they might affect the differing results. For instance, did study I and II use the same physicians? Was there a geographical/system/provider difference between the two studies which might cause differences in prescribing techniques? Was there a difference in missing data/quality of reported data between the two? Different inclusion criteria?

Reply: While the two studies were rather similar in design, they had distinct primary aims and were performed at different times. One of us (MCM) regularly teaches statistics courses and has been a regular statistical reviewer for J Urol. Based on this bona fide expertise, we wish to explain that a pooled analysis of two similar but distinct datasets is not the same as reporting both of them in parallel. While this creates added complexity, we consider it a key message of the manuscript that even under such similar conditions, two studies can yield different answers to the same question. We find this very helpful in understanding why previously reported studies for the very same questions (specific factors associated with dosing decisions) also reached heterogeneous answers. We had also discussed internally the option to report each study in a separate manuscript. However, given that the two did not reach the same conclusions on some questions, we felt that this would not only look as salami tactics but also generate impressions in one paper that were not confirmed in the other. Therefore, we would like to stick to our decision to report both studies separately.

Reviewer’s comment: ** I understand your rationale for this (I was considering perhaps a meta analysis rather than pooled data, although this might also be impractical with only two studies). I think it’s very important to note how very similar the studies were in design and population, and yet the differing outcomes, and I think it has important implications in comparison with other similar studies. I think it is worth addressing the decision to present both separately as a strength and a cautionary note to be circumspect when interpreting such studies alone.

Reply 2: That is a great suggestion. We have implemented this in a new paragraph of section 4.1 (critique of methods) in l. 350-359.

We are not sure what the referee means by “prescribing techniques”. While prescribing habits may have shifted over time, we have no evidence for differences in geographic representation. System/provider differences are not applicable to the German healthcare system. Moreover, we have also no evidence for a systematic difference in missing values between the studies. Finally, neither study had inclusion/exclusion criteria other than the Summary of Product Characteristics, which did not change between studies I and II.

Reviewer’s comment: **Thank you. I was particularly probing for potential systematic differences between the two studies which might explain outcome differences, related to my previous point. I think the reader might benefit from being explicitly told that you have not identified any differences which may affect study outcomes, and that prescribing recommendations/habits are reasonably homogeneous across the sample (e.g. per national guidelines).  

Reply 2: We have explicitly added such information to the new paragraph as part of section 4.1 mentioned above (l. 350-359).

Reviewer’s comment: This was confusing notation: ‘a reduction in urgency episodes by 4 (2; 7; 5.1± 4.6)’. Assuming the median is outside the parentheses and 2 is the lower quartile and 7 the UQ, please either write the IQR as a single number or a hyphenated range like 4(5; 5.1± 4.6) or 4(2-7; 5.1± 4.6).

Reply: We have adopted the suggestion and now consistently report IQR as ranges (e.g. 2-7). However, we would like to alert the reviewer to the fact that this leads to the strange writing of e.g. -5—1 in cases where the lower end is -5 and the upper end -1.

Reviewer’s comment: ** If the notation is confusing, your original notation could be used but please state that you provide the lower quartile; upper quartile rather than IQR to avoid confusion about the extra number.

Reply 2: For a statistician it is obvious that providing IQR means that lower and upper quartile are presented. Apparently, our assumption was wrong that clinical colleagues would see it the same way. Therefore, we have a) gone back to the original description with a ; separating lower and upper quartile to avoid confusions of double – and b) explained this in Methods (l. 127).

Reviewer’s comment: In 3.4, you state that greater improvement is associated with dose escalation, and that doesn’t seem to be the case – the smallest changes are in the escalation group.

Reply: There may be a misunderstanding: the table does not show changes but rather measured values at baseline and week 4. The legend of this table was revised for clarity, but the statements in the main text are correct.

Reviewer’s comment: **There is still a misunderstanding! You state in 3.4: “A greater number of urgency episodes at baseline and a greater improvement of incontinence episodes after 4 weeks of treatment were associated with dose escalation in both studies with a p < 0.05;” In study 1, the dose escalation group had 11 vs 9 urgency episodes in the non-escalation group, therefore greater urgency was associated with dose escalation. However, greater improvement in incontinence implies a greater reduction in incontinence episodes. Improvement in incontinence is not included in the table (this is why I suggested including change values in the tables, since you reference change values here). Looking at the graphs in figure 1, the dose escalation group had a reduction of 1 incontinence episode after 4 weeks, and the 30/30 group had a reduction of 2 episodes, which would suggest that dose escalation is associated with a smaller improvement in incontinence. In fact, in the discussion, you state “Adding to this heterogeneity in findings, dose-escalation was associated with greater baseline values and/or smaller initial improvements sometimes for all OAB symptoms, but sometimes such as also observed in the present study for only selected OAB symptoms.” Which, if I interpret this sentence correctly, seems to suggest that dose escalation in this study is associated with smaller initial improvements in some OAB symptoms. You also conclude that “Based on the overall evidence we conclude that both greater baseline OAB severity and smaller improvements upon initial treatment tend to be associated with a decision for dose escalation.” Both statements contradict the result you describe in 3.4. If the statement that greater improvements in incontinence is associated with dose escalation is indeed correct per the logistic regression model (although table 4 does not seem to suggest that), I think that this should be examined and discussed closely in the paper as it is counter-intuitive, both from the data provided and in practice. (This is also applicable to the improvement in number of nocturia episodes.)

Reply 2: We apologize for sloppy wording. Our models included data at baseline and after 4 weeks; changes/improvements were not direct parts of the models and only indirectly as difference between baseline and 4-week value. We adapted the corresponding wording to be more precise (l. 251, 252, 253, 264-265).

Reviewer’s comment: I think that the reader would benefit from some of the data tables in the supplement being in the actual paper. In fact, I think it would be much more helpful for the reader to have all OAB results in one place – it is very difficult to try to read some in the text and then compare to a table in the supplement.

Reply: We are open to this suggestion but would like to raise the issue that the referee proposed above the opposite, i.e. shifting more data from the main manuscript to the Online Supplement. The suggestion may have come from the fact that the tables intended for the main manuscript apparently got lost in the finally formatting steps. If the referee continues to recommend shifting information from the Online Supplement to the main manuscript after having seen the lost tables, we will be happy to implement that suggestion.

Reviewer’s comment: **I hope I was clearer in my intentions above – my suggestion of movement of means/SD to the supplement was a compromise if there was a space concern adding tables to the main paper. I certainly think the demographic table and current table 3 are useful additions to the main paper (I would also advocate for the corresponding study II table to be included). I am of the opinion that data tables with common summary statistics like this are of the most use to the reader and help the reader understand the paper. For the same reason, I appreciate your inclusion of means/SD. I don’t necessarily think that the tables of parameter estimates from the regressions need to be in the main paper if space is a concern, since these are less intuitive to the average clinical reader.

Reply 2: As the journal has no formal length limits and the editor did not make any suggestion on shortening, we saw no reason for shortening except in places where this made the story more readable. As explained above, Supplementary Tables 2 and 3 are identical to main Tables 1 and 3, except that the former show data from study II whereas the latter show data from study I. While the Supplementary Tables 2 and 3 are necessary for diligent reporting of the data from the second study, they do not add to the story. Therefore, we prefer to keep them in the Online Supplement. As improvements upon dose-escalation have repeatedly been shown for multiple muscarinic antagonists, we feel that the multiple regression models are the key pieces providing novel information in the present manuscript. We appreciate that they are more difficult to read for many clinical colleagues. Nonetheless, we would like to keep them in the main manuscript because this is where the main scientific novelty resides.

Reviewer’s comment: Since OAB parameters were associated with dose choice and escalation, baseline parameters are different between the groups. Might it therefore be helpful to consider a percentage improvement? Additionally, might it be worth considering the absolute value at week 4 as a predictor, assuming a physician might evaluate this against ‘tolerable’ ranges.

Reply: There are two ways the models can be set up to perform the analyses we intended. One way is to include absolute values at baseline and week 4, the other is to consider % changes. We had chosen the former approach because % changes already include baseline values in their calculations; as we also wanted to account for allocation bias, we had to include baseline values in the model. Had we included baseline and % changes, baseline would indirectly have entered the model twice, which would have contaminated the outcomes. Considering the absolute value at week 4 is what we have done (l. 153-154).

Reviewer’s comment: **The legend for table 4 suggests that “taking both OAB parameters after 4 and their change as compared to baseline into consideration” (I assume 4 weeks?), but the preceding results talks about urgency at baseline being associated. I had assumed that symptoms/24h in the table are baseline and ‘changes’ are taking both time points into account, but I think this needs a better explanation, since the legend suggests these are at the 4 week time point, in which case baseline values are not shown.

Reply 2: The same sloppy wording as above. We’re terribly sorry about this and happy to that you kept us straight. What was in the model (see l. 155-158) are baseline and 4-week values, not the changes. The left column of the table and the table legend were corrected (l. 258).

Reviewer’s comment: What do you mean by people who discontinued treatment ‘after the treatment period’? Were these people who completed the treatment but were lost to follow-up? It seems from the next sentence that all 145 discontinued during the treatment period. What was the breakdown of those who discontinued after visit 2 who had the escalated dose?

Reply: As our manuscript represents a post-hoc analysis of an existing database, the authors had no influence on how data were captured. While an observation period of about 12 weeks had been specified, the pre-specified statistical analysis plan counted patients stopping treatment after this as adverse drug reaction. This certainly is open to debate. However, it reflects that this was an observational study, i.e. nobody was treated for study purposes and only treatments initiated by the physician based on his decision were documented systematically; thus, stopping treatment after 12 weeks can be seen as adverse drug reaction. In any case, we as authors did not wish to down-size reported adverse drug reactions by changing how they were counted; most importantly, as an employee of the sponsoring company is an author, we want to avoid any suspicion that data on tolerability had been “manipulated”.

Reviewer’s comment: **I *think* I understand this. Please make this more clear in the text “those that discontinued drug upon conclusion of the observation period of this study are included in these numbers etc…”. The current sentence is still confusing.

Reply 2: We have added clarifying language in this regard (l. 310-311).

Reviewer’s comment: You state height as a variable associated with dose escalation, but since you did not test sex (I think you should be reporting sex, not gender, here) might height be a function of whether men are more likely to have dose escalation?

Reply: There may be a misunderstanding. Gender was included in all models; it is not reported in some outcome tables because it did not fulfill the criteria for retention in the model in some cases. Thus, parameter estimates for gender are included in Tables 2, 5 and 6.

Reviewer’s comment: ** Thank you for clarifying. Importantly, are you actually reporting gender (social/cultural roles/personal identity; man/woman/other) rather than sex (biological attributes; male/female)? Please confirm or switch ‘gender’ to ‘sex’.

Reply 2: Throughout the manuscript, we do not talk about sex. To clarify our use of the word gender, we have now defined it upon first mentioning (l. 61).

Round 3

Reviewer 2 Report

Thank you for your consideration of my comments. The additions have helped clarify my understanding of the paper.